# A 'double-edged' role for type-5 metabotropic glutamate receptors in pain disclosed by light-sensitive drugs

Serena Notartomaso[1†], Nico Antenucci[2†], Mariacristina Mazzitelli[2], Xavier Rovira[3], Serena Boccella[4], Flavia Ricciardi[4], Francesca Liberatore[1], Xavier Gomez-Santacana[3], Tiziana Imbriglio[1], Milena Cannella[1], Charleine Zussy[5], Livio Luongo[4], Sabatino Maione[4], Cyril Goudet[5], Giuseppe Battaglia[1,6], Amadeu Llebaria[3], Ferdinando Nicoletti[1,6]*, Volker Neugebauer[2,7,8]*

[1]Mediterranean Neurological Institute, IRCCS Neuromed, Pozzilli, Italy; [2]Department of Pharmacology and Neuroscience, Texas Tech University Health Sciences Center, Lubbock, United States; [3]MCS - Medicinal Chemistry & Synthesis, Institute for Advanced Chemistry of Catalonia, Barcelona, Spain; [4]Department of Experimental Medicine, Division of Pharmacology, University of Campania "Luigi Vanvitelli", Naples, Italy; [5]Institute of Functional Genomics IGF, National Centre for Scientific Research CNRS, INSERM, University of Montpellier, Montpellier, France; [6]Department of Physiology and Pharmacology, Sapienza University of Rome, Rome, Italy; [7]Center of Excellence for Translational Neuroscience and Therapeutics, Texas Tech University Health Sciences Center, Lubbock, United States; [8]Garrison Institute on Aging, Texas Tech University Health Sciences Center, Lubbock, United States

*For correspondence: ferdinandonicoletti@hotmail.com (FN); volker.neugebauer@ttuhsc.edu (VN)

[†]These authors contributed equally to this work

Competing interest: The authors declare that no competing interests exist.

**Abstract** We used light-sensitive drugs to identify the brain region-specific role of mGlu5 metabotropic glutamate receptors in the control of pain. Optical activation of systemic JF-NP-26, a caged, normally inactive, negative allosteric modulator (NAM) of mGlu5 receptors, in cingulate, prelimbic, and infralimbic cortices and thalamus inhibited neuropathic pain hypersensitivity. Systemic treatment of alloswitch-1, an intrinsically active mGlu5 receptor NAM, caused analgesia, and the effect was reversed by light-induced drug inactivation in the prelimbic and infralimbic cortices, and thalamus. This demonstrates that mGlu5 receptor blockade in the medial prefrontal cortex and thalamus is both sufficient and necessary for the analgesic activity of mGlu5 receptor antagonists. Surprisingly, when the light was delivered in the basolateral amygdala, local activation of systemic JF-NP-26 reduced pain thresholds, whereas inactivation of alloswitch-1 enhanced analgesia. Electro-physiological analysis showed that alloswitch-1 increased excitatory synaptic responses in prelimbic pyramidal neurons evoked by stimulation of presumed BLA input, and decreased BLA-driven feed-forward inhibition of amygdala output neurons. Both effects were reversed by optical silencing and reinstated by optical reactivation of alloswitch-1. These findings demonstrate for the first time that the action of mGlu5 receptors in the pain neuraxis is not homogenous, and suggest that blockade of mGlu5 receptors in the BLA may limit the overall analgesic activity of mGlu5 receptor antagonists. This could explain the suboptimal effect of mGlu5 NAMs on pain in human studies and validate photopharmacology as an important tool to determine ideal target sites for systemic drugs.

## Editor's evaluation

In this convincing study, the authors have used light-sensitive mGlu5 negative allosteric modulators to determine the role of these receptors in a chronic pain model. These findings are have important

implications for the pain field, and will generally valuable to neuroscientists interested in signaling through mGluR5 receptors.

## Introduction

Light-based tools have transformed modern biology. Beyond the established biomolecular detection and visualization technologies (*Rodriguez et al., 2017*; *Wu and Shroff, 2022*), recent developments use light as a control element to direct the activity of cells and biomolecules in vitro and in vivo (*Szobota and Isacoff, 2010*). The strategic introduction of chromophores as light-responsive molecular elements in cells has resulted in a precise regulation of biological function with high spatial, temporal, and biochemical precision. After light absorption, the chromophore undergoes chemical changes that modify the structure and function of the receptor and can be reversibly activated and deactivated.

Photopharmacology (or optopharmacology) is an innovative technique that combines the use of light and photoresponsive molecules to obtain precise spatiotemporal control of drug activity (*Paoletti et al., 2019*; *Hüll et al., 2018*; *Frank et al., 2015*; *López-Cano et al., 2023*). As such, in vivo photopharmacology has emerged as a unique approach to identifying mechanisms of drug actions and as the doorway to precision medicine (*Hüll et al., 2018*; *Fuchter, 2020*; *Velema et al., 2014*). Using light-sensitive drugs, it is possible to optimize the efficacy of new therapeutic agents and minimize their adverse effects resulting from widespread actions after systemic administration. The use of photopharmacology-based approaches to interrogate receptors in the brain is attractive due to the possibilities of manipulating the activity of specific localizations during defined times (*Kramer et al., 2013*; *Matera and Bregestovski, 2022*).

Photopharmacological approaches have recently been explored for pain modulation by members of our team (*Font et al., 2017*; *Notartomaso et al., 2022*; *Pereira et al., 2023*) and by others (*Ma et al., 2023*; *Landra-Willm et al., 2023*). The drug treatment of neuropathic pain, i.e., chronic pain originating from damage or disease of the somatosensory system, remains a major therapeutic challenge and unmet medical need because of severe side effects and a high percentage of patients resistant to medication (*Fornasari, 2017*; *Finnerup et al., 2015*). Neuropathic pain reflects the development of neuroplasticity that drives nociceptive sensitization, i.e., the amplification of pain signal transmission in most stations of the pain neuraxis, including regions of the pain matrix that encode the perceptive and emotional-affective components of pain (*Latremoliere and Woolf, 2009*; *Carozzi et al., 2008*; *Colloca et al., 2017*). Targeting glutamate receptors represents a potential strategy aimed at restraining the maladaptive form of synaptic plasticity associated with chronic pain, and photomolecular tools directed at glutamate (*Levitz et al., 2017*) receptors have been developed combining chemical and genetic techniques.

A large body of evidence suggests that type-5 metabotropic glutamate (mGlu5) receptors are candidate drug targets for the treatment of neuropathic pain. mGlu5 receptors, which belong to group-I mGlu receptors, are coupled to $G_{q/11}$ proteins, and their activation leads to polyphosphoinositide (PI) hydrolysis, with the ensuing formation of inositol-1,4,5-trisphosphate and diacylglycerol (*Nicoletti et al., 2011*). mGlu5 receptors are widely expressed in most regions of the pain neuraxis and can contribute to the development of nociceptive sensitization. Pharmacological blockade of mGlu5 receptors has mostly shown antinociceptive activity in preclinical models of inflammatory and neuropathic pain (*Mazzitelli et al., 2022*; *Nicoletti et al., 2015*). In contrast, oral administration of a mGlu5 receptor-negative allosteric modulator (NAM, fenobam), had minimal temporary analgesic effects and no clinically or statistically significant persistent analgesic effect on heat/capsaicin-induced cutaneous hyperalgesia in healthy volunteers (*Cavallone et al., 2020*). It remains to be determined if, and to what extent, region-specific actions are involved in the analgesic activity of mGlu5 receptor NAMs, considering that opposing actions have been reported (*Chung et al., 2018*; *Kiritoshi et al., 2016*).

To address this question, we applied photopharmacology to identify brain region-specific functions of mGlu5 in neuropathic pain, using the well-established chronic constriction injury (CCI) model (*Bennett and Xie, 1988*). We designed a new experimental strategy that combined the systemic application of photocaged, or photoswitchable compounds with their optical activation, silencing or reactivation in specific brain regions. We tested two functionally distinct light-sensitive mGlu5 receptor

ligands: (i) a caged derivative of the mGlu5 receptor NAM raseglurant (compound JF-NP-26), which is inactive on its own and activated by light in the visible spectrum (405 nm) *Font et al., 2017*; and, (ii) the photoswitchable mGlu5 receptor NAM alloswitch-1, which is active on its own, is inactivated by light at 405 nm, and can be re-activated by light at 520 nm (*Pittolo et al., 2014*).

The use of light allows a highly restricted spatio-temporal control of drug activity without genetic manipulation of targeted receptor proteins (as opposed to optogenetics). In addition, light in the visible spectrum does not damage CNS tissue and can be delivered multiple times if needed. The use of photoactive drugs also allows an extremely rapid control of pain, circumventing the anatomical barrier that limits the access of analgesic agents to their target sites. For example, light-induced activation of JF-NP-26 in the thalamus caused rapid and robust analgesia in a mouse model of breakthrough cancer pain, a sudden severe, and transient type of pain that may develop in cancer patients under treatment with opioids (*Notartomaso et al., 2022*). Light-activation of JF-NP-26 can help identify the site within the pain neuraxis where mGlu5 receptor blockade is *sufficient* to cause analgesia. In contrast, light-inactivation of alloswitch-1 allows to establish the region where mGlu5 receptor blockade is *necessary* for the induction of analgesia. In combination with electrophysiological analyses, the photopharmacological approach can shed light on the functional circuit controlling pain and other nervous system functions in response to the activation of mGlu5 receptors.

For brain region-specific optical activation and inactivation of mGlu5 NAMs to determine the site of action of mGlu5 receptors, we selected prefrontal cortical (infralimbic, prelimbic and anterior cingulate cortices) and subcortical (thalamus and amygdala) regions that have been implicated in sensory, affective and cognitive aspects of pain and pain modulation (*Colloca et al., 2017*; *Neugebauer et al., 2023*). This study shows that prefrontal cortical regions and thalamus are necessary and sufficient for the antinociceptive effects of mGlu5 receptor blockade by systemic NAMs in neuropathic pain conditions. Unexpectedly, however, local mGlu5 receptor inhibition in the basolateral amygdala produced pronociceptive rather than antinociceptive effects by blocking feedforward inhibition of the central nucleus of amygdala without affecting amygdala output to prelimbic cortex (see Figure 4). Thus, the use of light-sensitive drugs allowed the demonstration of a heterogenous function of mGlu5 receptors in the pain neuraxis, which may have important implications for the application of precision medicine in the treatment of chronic pain.

## Results

The aim of this study was to determine the region-specific behavioral and electrophysiological effects of mGlu5 receptor blockade with caged and photoswitchable NAMs in neuropathic pain using photopharmacology. While the effect of optical manipulations of photosensitive mGlu5 NAMs in individual brain regions in pain models has been studied before (*Font et al., 2017*; *Notartomaso et al., 2022*; *Gómez-Santacana et al., 2017*), the comparative analysis across different brain regions and their contribution to systemically applied NAMs represents a novel concept.

### mGlu5 receptor signaling in brain regions of the pain matrix in neuropathic pain

Mice with chronic constriction injury (CCI) of the left sciatic nerve showed the expected hypersensitivity reflected in the decrease of mechanical pain thresholds measured in the ipsilateral hind paw after 16 days (*Figure 1a and b*). We examined mGlu5 receptor signaling in regions of the contralateral pain matrix using an in vivo method for the assessment of mGlu5-receptor-mediated PI hydrolysis (*Zuena et al., 2018*). Mice were pretreated with lithium ions and then challenged with a selective mGlu5 receptor-positive allosteric modulator (PAM, VU0360172, 30 mg/kg, i.p.). mGlu5-mediated PI hydrolysis (i.e. the extent of VU0360172-stimulated InsP formation) was amplified in the contralateral infralimbic cortex, prelimbic cortex, anterior cingulate cortex, and basolateral amygdala (BLA), but not in the contralateral ventrobasal thalamus (VPL), of CCI mice compared to the corresponding regions of sham-operated mice (*Figure 1c–g*). mGlu5 receptor protein levels were up-regulated in the contralateral infralimbic cortex, but not in other brain regions, of CCI mice (*Figure 1—figure supplement 1*). The findings suggest that mGlu5 receptors become hyperactive at least in four regions of the pain matrix after induction of neuropathic pain.

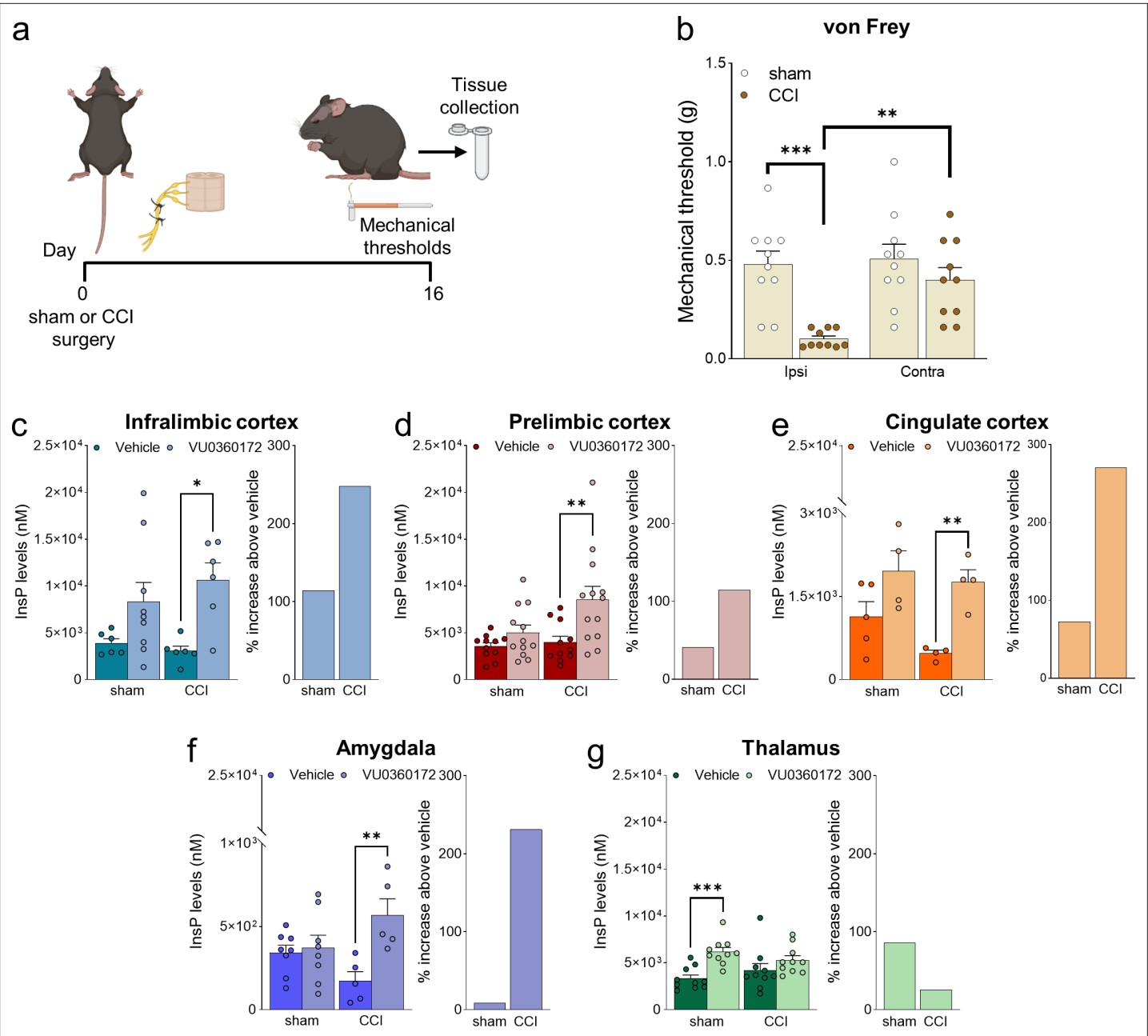

**Figure 1.** Neuropathic pain model and mGlu5 receptor signaling in different brain regions. (**a**) Experimental protocol. (**b**) Mechanical (hyper-)sensitivity on the left hind paw measured with von Frey filaments was reduced in chronic constriction injury (CCI) mice (n=10) versus sham-operated mice (n=10) 16 days after nerve ligation. Bar histograms show means ± SEM. (**c–g**) mGlu5 receptor-mediated polyphosphoinositide (PI) hydrolysis in different brain regions of sham and CCI mice. Increased InsP levels were observed in infralimbic (**c**), prelimbic (p=0.051) (**d**), and cingulate (**e**) cortices and amygdala (**f**), but not thalamus (**g**), of neuropathic mice (CCI) after stimulation with a positive allosteric modulator (PAM, VU0360172, 30 mg/kg i.p.) compared to vehicle. Enhanced InsP concentration was detected in the thalamus, but not in the prefrontal cortical regions and amygdala, of sham animals treated with VU0360172 compared to vehicle. Bar histograms on the right in c-g show a percent increase above vehicle in sham and CCI mice. Bar histograms show means ± SEM of 10 (**b**), 6–9 (**c**), 11-13(d), 4–5 (**e**), 5–8 (**f**), 10 (**g**) mice per group. Two-way ANOVA: (**b**) sham vs CCI, $F_{(1,36)}$=16.04; p=0.0003; ipsi vs contra, $F_{(1,36)}$=7.304; p=0.0104; interaction $F_{(1,36)}$=5.062, p=0.0307; (**c**) sham vs CCI, $F_{(1,23)}$=0.1952; p=0.6627; vehicle vs VU0360172, $F_{(1,23)}$=12.78, p=0.0016; interaction $F_{(1,23)}$=0.8758, p=0.3591; (**d**) sham vs CCI, $F_{(1,43)}$=4.421; p=0.0414; vehicle vs VU0360172, $F_{(1,43)}$=10.15, p=0.0027; interaction, $F_{(1,43)}$=2.685, p=0.1086; (**e**) sham vs CCI, $F_{(1,13)}$=2.833; p=11.62; vehicle vs VU0360172, $F_{(1,13)}$=16.67, p=0.0013; interaction $F_{(1,13)}$=0.7931, p=0.3893; (**f**) sham vs CCI, $F_{(1,22)}$=0.03868, p=0.8459; vehicle vs VU0360172, $F_{(1,22)}$=9.039, p=0.00065; interaction $F_{(1,22)}$=6.679, p=0.0169; (**g**) sham vs CCI, $F_{(1,36)}$=1.153e-005, p$P$=0.9973; vehicle vs VU0360172, $F_{(1,36)}$=14.76, p=0.0005; interaction $F_{(1,36)}$=3.062, p=0.0887. *p<0.05, **p<0.01; ***p<0.001, Bonferroni's multiple comparison post hoc test. Source files are available in the ***Figure 1—source data 1***.

*Figure 1 continued on next page*

*Figure 1 continued*

The online version of this article includes the following source data and figure supplement(s) for figure 1:

**Source data 1.** Neuropathic pain model and mGlu5 receptor signaling in different brain regions.

**Figure supplement 1.** mGlu5 receptor protein levels in different brain regions in sham and chronic constriction injury (CCI) mice.

**Figure supplement 1—source data 1.** mGlu5 receptor protein levels in different brain regions in sham and CCI mice.

## Optical modulation of mGlu5 receptors

We applied photopharmacology to study brain region-specific functions and pharmacological manipulations of mGlu5 receptors by implanting optic fibers in the contralateral infralimbic cortex, prelimbic cortex, anterior cingulate cortex, amygdala (BLA), and thalamus (VPL) of CCI and sham mice 12 days after surgery (*Figure 2a and b*). Four days later, mice received a systemic injection of JF-NP-26 (10 mg/kg, i.p.), alloswitch-1 (10 mg/kg, i.p.), or their respective vehicle (*Figure 2b, i and o*). JF-NP-26 is an inactive pro-drug of the mGlu5 receptor NAM raseglurant and can be activated by blue-violet light (405 nm) (*Font et al., 2017*). The *trans*-isomer of alloswitch-1 is a systemically active mGlu5 receptor NAM, which can be isomerized into inactive *cis*-alloswitch-1 by blue-violet light (405 nm) and converted back into the active *trans*-isomer by green light (520 nm) (*Pittolo et al., 2014*; *Figure 2c*).

## Behavioral effects of brain region-specific activation of systemic mGlu5 NAM JF-NP-26

Light-induced activation of JF-NP-26 in the infralimbic cortex, prelimbic cortex, anterior cingulate cortex, or thalamus (VPL) contralateral to CCI caused rapid and robust antinociception in neuropathic mice measured as increased mechanical thresholds, whereas, unexpectedly, JF-NP-26 activation in the amygdala (BLA) further reduced mechanical pain thresholds, indicating pronociceptive effects (*Figure 2d–h*). JF-NP-26 (10 m/kg, i.p.) was given systemically 25 min before light-induced activation (*Figure 2i*). Interestingly, light-induced activation of JF-NP-26 in the prelimbic cortex contralateral to CCI also enhanced the mechanical pain threshold in the uninjured hind limb (ipsilateral to light delivery), whereas hyperalgesia of the uninjured limb was induced by JF-NP-26 activation in the BLA contralateral to CCI (*Figure 2—figure supplement 1*). No changes in mechanical thresholds were induced by the activation of JF-NP-26 in any brain region of sham-operated mice (*Figure 2—figure supplement 2*).

The data suggest that mGlu5 receptor blockade in medial prefrontal cortical regions (infralimbic, prelimbic, or anterior cingulate cortices) or thalamus, but not amygdala, is sufficient for antinociceptive effects in neuropathic pain. Knowing that mGlu5 receptors in the BLA shape susceptibility to stress and fear in rodents (*Shallcross et al., 2021*; *Kim et al., 2023*), we also measured depression-like and risk-taking behavior after light-induced activation of JF-NP-26 in the BLA of neuropathic mice. Light-induced activation of JF-NP-26 decreased risk-taking hence increased anxiety-like behavior in CCI mice as shown by the decreased number of entries into, and reduced time spent in, the light compartment of the light-dark box (*Figure 2—figure supplement 3a–c*). Depression-like behavior assessed with the tail-suspension test was unchanged in CCI mice after light-induced irradiation of JF-NP-26 in the BLA (*Figure 2—figure supplement 3d*).

## Behavioral effects of brain region-specific inactivation/reactivation of systemic mGlu5 NAM alloswitch-1

We used optical inactivation and reactivation of alloswitch-1 to determine the brain region(s) necessary for analgesia caused by systemic blockade of mGlu5 receptors with alloswitch-1. Systemic injection of alloswitch-1 (10 mg/kg, i.p.) consistently increased mechanical pain thresholds of CCI mice, indicating antinociceptive effects. Blue-violet-light-induced inactivation of alloswitch-1 in the contralateral infralimbic cortex, prelimbic cortex, and thalamus (VPL) abolished alloswitch-1-induced antinociception, which was restored by green-light induced reactivation of alloswitch-1 (*Figure 2j, k and n*). In contrast, optical inactivation of alloswitch-1 in the amygdala (BLA) further increased the antinociceptive effects of systemic alloswitch-1, and this effect was reversed by reactivation of alloswitch-1 (*Figure 2m*). Optical modulation of alloswitch-1 in the anterior cingulate cortex had no effect on pain thresholds (*Figure 2l*). Alloswitch-1 was given systemically 25 min before light-induced inactivation (*Figure 2o*). The data suggest that mGlu5 receptor blockade in infralimbic and prelimbic cortices and

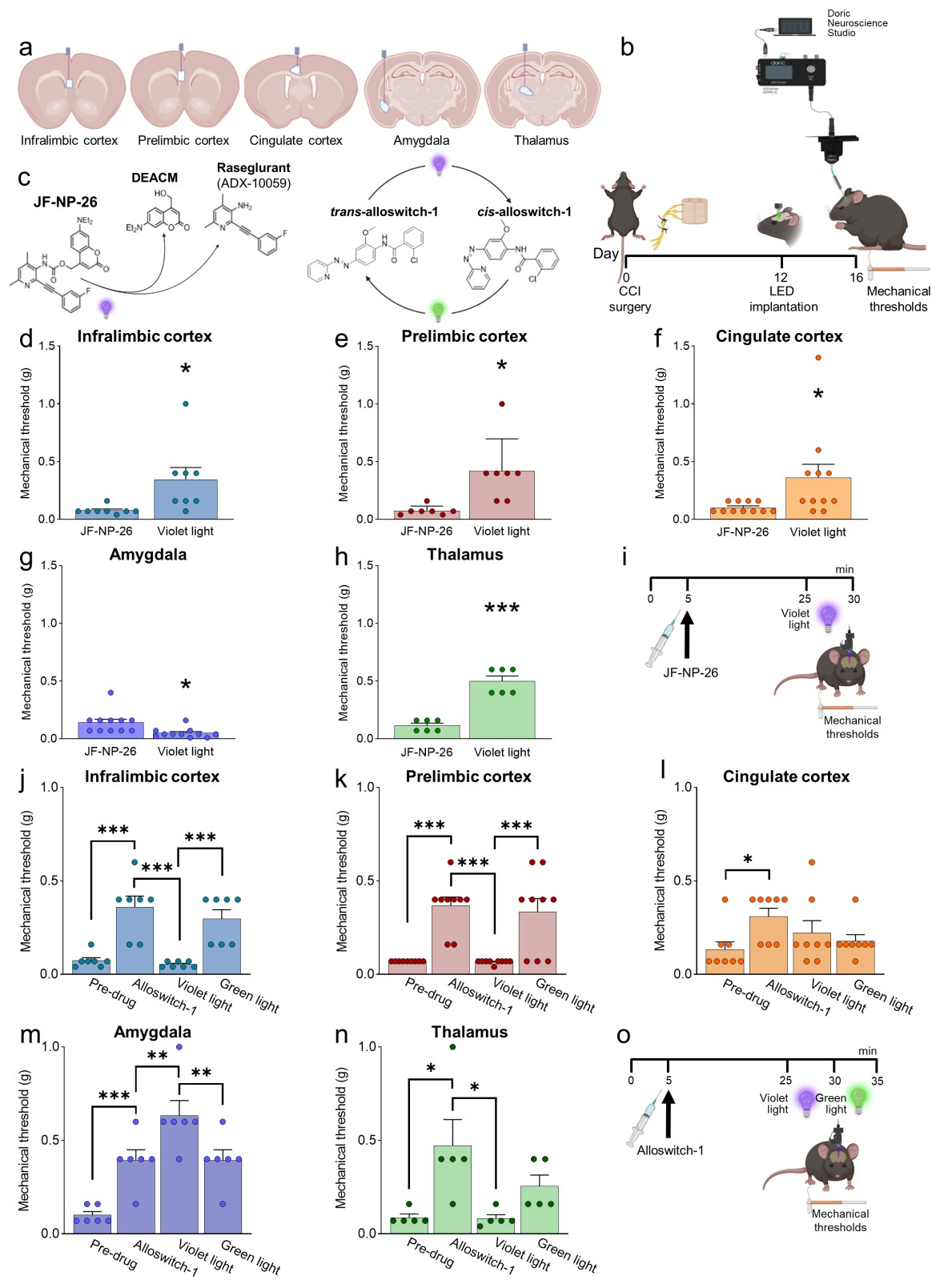

**Figure 2.** Behavioral effects of light-induced manipulations of mGlu5 receptors in different brain regions in chronic constriction injury (CCI) mice. Schematic representation of sites of stereotaxic implantation of LED optical fibers in different brain regions. (**b**) Experimental protocol for optical modulation. (**c**) Optical activation (blue-violet light, 405 nm) of mGlu5 NAM JF-NP-26 and optical inactivation (blue-violet light, 405 nm) and reactivation (green light, 520 nm) of mGlu5 NAM alloswitch. (**d-i**) Optical activation (blue-violet light, 405 nm) of mGlu5 NAM JF-NP-26 (10 mg/kg, i.p.) in

*Figure 2 continued on next page*

*Figure 2 continued*

contralateral (to the side of CCI) infralimbic cortex (**d**), prelimbic cortex (**e**), anterior cingulate cortex (**f**), and thalamus (**g**) caused a significant increase in mechanical thresholds, whereas JF-NP-26 activation in the amygdala (**h**) caused hyperalgesia, in neuropathic mice 16 days after CCI induction (**i**). (**j–o**) Systemic application of mGlu5 NAM alloswitch-1 (10 mg/kg, i.p.) caused antinociception in CCI mice, compared to vehicle. Optical inactivation (blue-violet light, 405 nm) of alloswitch-1 in the contralateral (to the side of CCI) infralimbic cortex (**j**), prelimbic cortex (**k**), or thalamus (**n**) reversed the antinociceptive effects, while light-induced (green light, 520 nm) reactivation of alloswitch-1 in those brain regions reinstated analgesia. No significant behavioral changes were observed with optical manipulations in the anterior cingulate cortex (**l**). In contrast, optical inactivation of alloswitch-1 in the amygdala with blue-violet light further increased mechanical thresholds (enhancement of antinociception), and this effect was reversed by reactivation of alloswitch-1 with green light (**m**). Bar histograms show mean ± SEM of 8 (**d**), 7 (**e**), 11 (**f** and **g**), 6 (**h**), 7 (**j**), 9 (**k**), 8 (**l**), 6 (**m**), and 5 (**n**) mice per group. (**d-h**) *p<0.05, ***p<0.001, paired student's t-test compared to JF-NP-26 without light activation. (**d**) t=2.487, p=0.0418; (**e**) t=3.243, p=0.0176; (**f**) t=2.390, p=0.0379; (**g**) t=3.025, p=0.0128; (**h**) t=9.062, p=0.0003. (**j–n**) One-way repeated measures ANOVA; (**j**) F (3,18)=19.25, p<0.0001; (**k**) F (3,24)=20.12, p<0.0001; (**l**) F (3,21)=3.813, p=0.0251; (**m**) F (3,15)=24.66, p<0.0001; (**n**) F (3,12)=5.463, p=0.0133. *p<0.05, **p<0.01, ***p<0.001, Bonferroni's multiple comparisons post hoc tests. Source files are available in the *Figure 2—source data 1*.

The online version of this article includes the following source data and figure supplement(s) for figure 2:

**Source data 1.** Behavioral effects of light-induced manipulations of mGlu5 receptors in different brain regions in CCI mice.

**Figure supplement 1.** Effects of light-induced blockade of mGlu5 receptors in different brain regions on mechanical pain thresholds in the unlesioned paw of chronic constriction injury (CCI) mice.

**Figure supplement 1—source data 1.** Effects of light-induced blockade of mGlu5 receptors in different brain regions on mechanical pain thresholds in the unlesioned paw of CCI mice.

**Figure supplement 2.** Effects of light-induced blockade of mGlu5 receptors in different brain regions on mechanical pain thresholds in the left paw of sham mice.

**Figure supplement 2—source data 1.** Effects of light-induced blockade of mGlu5 receptors in different brain regions on mechanical pain thresholds in the left paw of sham mice.

**Figure supplement 3.** Effects of light-induced blockade of mGlu5 receptors in the right basolateral amygdala (BLA) on anxiety-like (risk-taking) and depressive-like behaviors of sham and chronic constriction injury (CCI) mice.

**Figure supplement 3—source data 1.** Effects on risk-taking and depressive-like behavior of sham and CCI mice.

thalamus, but not anterior cingulate cortex, is necessary for antinociceptive effects in neuropathic pain conditions, whereas mGlu5 blockade in the BLA has paradoxical pain-facilitating effects.

## Neural circuitry underlying the antinociceptive effects of mGlu5 blockade in the prelimbic cortex

To understand the neural basis for the antinociceptive behavioral effects of optical modulation of alloswitch-1 in the medial prefrontal cortex, we conducted whole-cell patch-clamp electrophysiological recordings from pyramidal neurons in layer 5 of the prelimbic cortex in brain slices from CCI mice 16 days post-injury. Excitatory and inhibitory postsynaptic currents (EPSCs and IPSCs) were evoked in visually identified pyramidal neurons by stimulation of presumed afferents from BLA that traverse the infralimbic cortex in layer 4 (*Figure 3a–c*), as described previously (*Ji et al., 2010*). EPSCs and IPSCs were blocked by bath application of a glutamate receptor antagonist (CNQX, 20 µM) confirming glutamatergic EPSCs and glutamate-driven IPSCs that are part of a feedforward inhibitory circuitry from the BLA to regulate prefrontal cortical output (*Neugebauer et al., 2023*; *Ji et al., 2010*) (IPSCs were additionally blocked by bicuculline, not shown). Synaptic responses were measured before (in artificial cerebrospinal fluid, ASCF) and during the application of alloswitch-1 to the brain slice, and during blue-violet and green light illumination to inactivate and reactivate, respectively, alloswitch-1 (*Figure 3d and e*).

Administration of mGlu5 NAM alloswitch-1 (100 nM) increased EPSCs compared to predrug ACSF values. Alloswitch-1 inactivation with blue-violet light (5 min, 0.5 Hz, 500 ms) slightly decreased EPSCs so that they were not significantly different from baseline values. Reactivation of alloswitch-1 with green light (5 min, 0.5 Hz, 500 ms) led to a significant increase in EPSCs compared to predrug (*Figure 3d and f*). No significant effects were observed on IPSCs (*Figure 3e and g*). As a result, the E/I ratio showed a pattern that reflected the changes in EPSCs with increases by alloswitch-1 and by reactivation with green light, and no change from baseline during blue-violet light application to inactivate alloswitch-1 (*Figure 3h*). These results suggest that mGlu5 receptor controls the excitatory drive of prelimbic output neurons, possibly by activation of synaptic inhibition of excitatory inputs; and mGlu5 receptor inhibition restores the excitatory drive-in neuropathic pain.

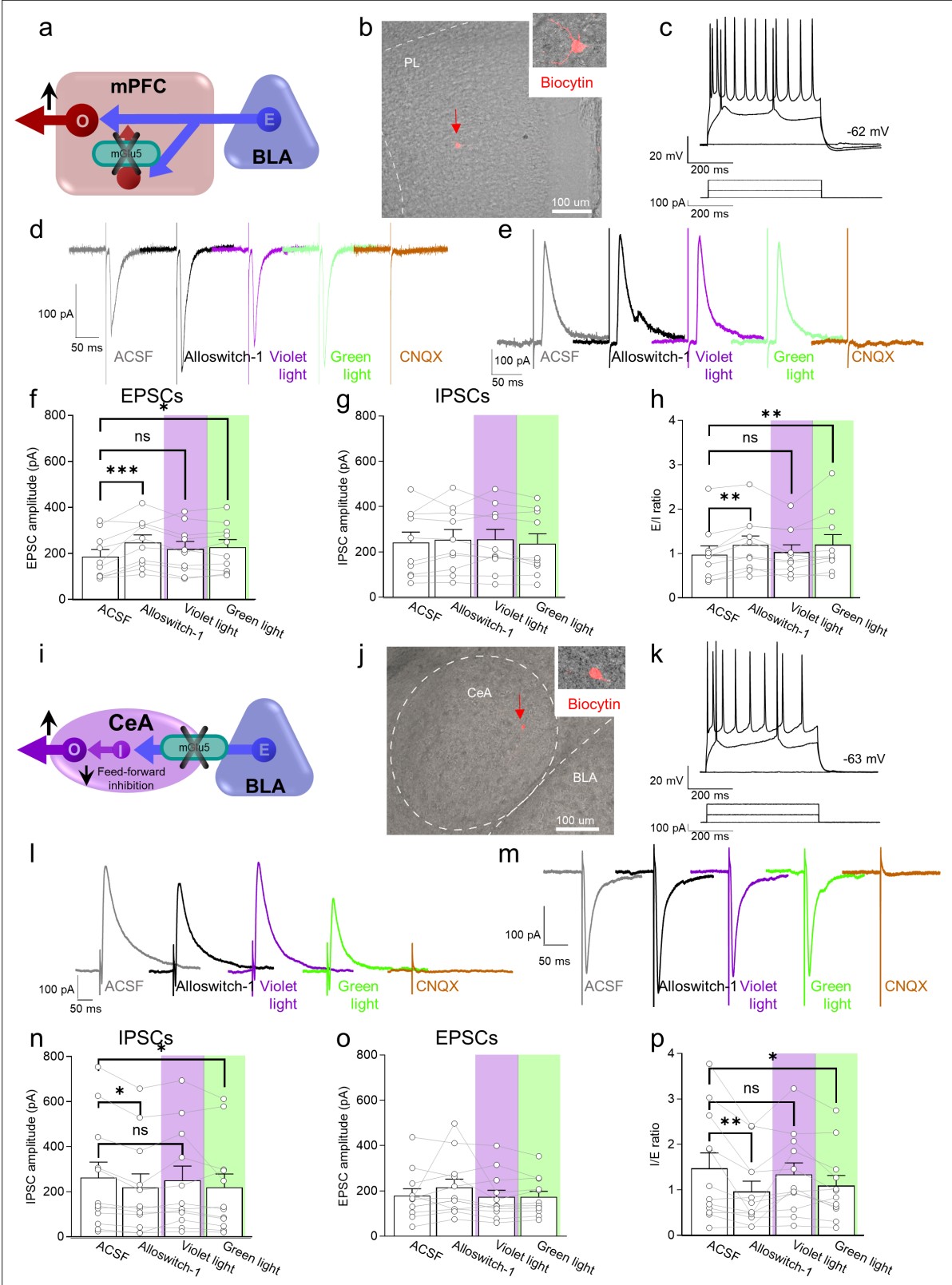

**Figure 3.** Electrophysiological effects of light-induced manipulations of mGlu5 on prelimbic pyramidal neurons and on amygdala feed-forward inhibition in neuropathic pain. (**a**) Prelimbic circuitry to explain mGlu5 receptor action. Basolateral amygdala (BLA) input (**E**) activates (**I**) interneuron projects onto excitatory (**E**) inputs to pyramidal output neurons (**O**). (**b, c**) Whole-cell patch-clamp electrophysiological recordings were performed to visually identified layer 5 pyramidal neurons in brain slices obtained from CCI mice 16 days after induction. Traces recorded in an individual neuron

*Figure 3 continued on next page*

*Figure 3 continued*

show typical regular action potential firing patterns in response to intracellular depolarizing current injections. (**d, e**) Excitatory postsynaptic currents (Excitatory postsynaptic currents (EPSCs), recorded at –70 mV; **d**) and inhibitory postsynaptic currents (inhibitory postsynaptic currents (IPSCs), recorded at 0 mV; **e**) were evoked by focal electrical stimulation of fibers of passage in the infralimbic cortex. EPSCs and IPSCs were blocked by a glutamate receptor antagonist (CNQX, 20 µM) confirming glutamatergic EPSCs and glutamate-driven IPSCs. Synaptic responses were evaluated before (ASCF) and during alloswitch-1 bath application, and under blue-violet and green light illumination. (**f**) Alloswitch-1 (100 nM by superfusion) enhanced the peak amplitude of EPSCs. (**g**) EPSCs were not different from baseline with blue-violet light-induced inactivation of alloswitch-1 (5 min, 0.5 Hz, 500 ms) while subsequent reactivation under green light illumination (5 min, 0.5 Hz, 500 ms) significantly increased EPSCs compared to pre-drug values. (**g**) No significant changes were observed on the IPSCs. (**h**) Changes in excitatory/inhibitory (E/I) ratio. (**i**) Amygdala circuitry to explain mGlu5 receptor action. Excitatory (**E**) input from the basolateral to central nucleus engages inhibitory (**I**) signaling for feedforward inhibition of output (**O**) neurons. (**j, k**) Whole-cell patch-clamp electrophysiological recordings were performed from amygdala neurons (latero-capsular division, CeLC) of brain slices obtained from CCI mice 16 days after induction. (**l, m**) Glutamate-driven IPSCs (recorded at 0 mV; **l**) and monosynaptic EPSCs (recorded at –70 mV; IPSCs and EPSCs were blocked by CNQX, 20 µM, confirming glutamate-driven IPSCs (feedforward inhibition) and glutamatergic EPSCs; **m**) were evoked in CeLC neurons by focal electrical stimulation in the BLA. Synaptic responses were evaluated before (ASCF) and during alloswitch-1 bath application, and under blue-violet and green light illumination. (**n**) Alloswitch-1 (100 nM) significantly decreased the peak amplitude of the IPSCs; IPSCs were not significantly different from baseline with blue-violet light-induced inactivation of alloswitch-1 (5 min, 0.5 Hz, 500 ms), whereas subsequent drug reactivation under green light illumination (5 min, 0.5 Hz, 500 ms) significantly decreased IPSCs compared to pre-drug values. (**o**) No significant changes were observed on the EPSCs. (**p**) Changes in inhibitory/excitatory (I/E) ratio. The data suggest that mGlu5 blockade in the CeA reduced BLA-driven feed-forward inhibition onto the CeLC neurons resulting in behavioral hypersensitivity. Bar histograms show mean ± SEM of n=10 in 7 mice (**f, g, h**) and n=12 in 8 mice (**n, o, p**) neurons. One-way ANOVA repeated measures: (**f**) $F_{(3,27)}=6.323$, p=0.0022; (**g**) $F_{(3,27)}=0.9374$, p=0.4362; (**h**) $F_{(3,27)}=5.499$, p=0.0044; (**n**) $F_{(3,33)}=4.079$, p=0.0144; (**o**) $F_{(3,33)}=1.779$, p=0.1704; (**p**) $F_{(3,33)}=5.581$, p=0.0033. Dunnett's multiple comparisons post hoc test. *p<0.05; **p<0.01 compared to pre-drug. Source files are available in the *Figure 3—source data 1*.

The online version of this article includes the following source data and figure supplement(s) for figure 3:

**Source data 1.** Electrophysiological effects of light-induced manipulations of mGlu5 on prelimbic pyramidal neurons and on amygdala feed-forward inhibition in neuropathic pain.

**Figure supplement 1.** Effects of light-induced blockade of mGlu5 receptors in the prelimbic cortex on descending pain control (rostroventromedial medulla, RVM) and in the basolateral amygdala (BLA)-prelimbic circuitry in chronic constriction injury (CCI) mice.

**Figure supplement 1—source data 1.** Effects of light-induced blockade of mGlu5 receptors in prelimbic cortex on descending pain control (RVM) and on BLA-prelimbic circuitry in CCI mice.

To explore if the antinociceptive behavioral effects of mGlu5 receptor inhibition in the prelimbic cortex could be explained by engaging descending pain control systems as a consequence of increased excitatory drive and output, we performed in vivo single-unit recordings of so-called ON-cells in the rostroventromedial medulla (RVM). RVM ON-cells are a critical element of top-down control of pain modulation and serve pronociceptive functions (*Chen and Heinricher, 2022*). Systemic administration of alloswitch-1 (10 mg/kg, i.p.) caused a trend towards decreased excitation of ON cells. Optical inactivation of alloswitch-1 in the prelimbic cortex increased ON-cell excitation whereas reactivation of alloswitch-1 with green light decreased excitation (*Figure 3—figure supplement 1*). The effect size of optical modulation in the prelimbic cortex on RVM activity, according to Cohen's *d* calculation from t-tests, is shown in the Table 1 (see *Supplementary file 1*).

These results suggest that increased prelimbic cortical output with mGlu receptor inhibition (brain slice physiology data) can decrease excitation of pronociceptive RVM ON-cells resulting in antinociceptive behavioral effects (see *Figure 3—figure supplement 1*). These findings highlight the importance of prefrontal cortical-RVM connectivity in mediating the pain-relieving effects associated with mGlu5 receptor inhibition in the prelimbic cortex.

## Neural circuitry underlying the paradoxical pronociceptive effects of mGlu5 blockade in the amygdala

We initially hypothesized that the neural basis for the pronociceptive behavioral effects of optical modulation of alloswitch-1 in the BLA could be the decrease of excitatory drive from BLA to the prefrontal cortex. Our previous work showed increased feedforward inhibition of prefrontal cortex due to increased activity in BLA in pain, and therefore, restoring prefrontal cortical output was able to inhibit pain behaviors (*Neugebauer et al., 2023*). We conducted in vivo electrophysiological recordings of individual pyramidal neurons in the prelimbic cortex in anesthetized mice 16 days after CCI induction. Systemically applied alloswitch-1 (10 mg/kg, i.p.) increased neuronal activity, but inactivation of alloswitch-1 with blue-violet light illumination in the BLA had no significant effect, arguing

against a contribution of mGlu5 in the BLA. Furthermore, reactivation of alloswitch-1 with green light illumination in the BLA did not mimic the effect of systemic alloswitch-1 (*Figure 3—figure supplement 1*). Therefore, we explored another BLA output and tested the hypothesis that inhibition of mGlu5 in the BLA would decrease feedforward inhibition of neurons in the central nucleus (CeA) and hence increase their activity (*Figure 3i*), which has been shown to correlate positively with pain behaviors through projections to several brain areas involved in behavioral modulation (*Neugebauer, 2020*).

To do so we performed whole-cell patch-clamp recordings of neurons in the laterocapsular division of the central nucleus of the amygdala (CeLC) in brain slices obtained from neuropathic mice 16 days after CCI induction (*Figure 3j and k*). CeA neurons receive direct excitatory and feed-forward inhibitory inputs from the BLA (*Neugebauer, 2020*). Glutamatergic-driven IPSCs and monosynaptic EPSCs were evoked in CeLC neurons by focal electrical stimulation in the BLA as described previously (*Ren et al., 2013*). Synaptic responses were evaluated before (ACSF) and during alloswitch-1 bath application, and under blue-violet or green light illumination to block and reinstate, respectively, the pharmacological activity of alloswitch-1 (*Figure 3l and m*). Both EPSCs and IPSCs were blocked by a glutamate receptor antagonist (CNQX, 20 µM), confirming glutamatergic EPSCs and glutamate-driven IPSCs (IPSCs were additionally blocked by bicuculline, not shown). Alloswitch-1 (100 nM) had significant inhibitory effects on IPSCs at the BLA-CeLC synapse (*Figure 3l and n*), which was reversed by inactivation with blue-violet light (5 min, 0.5 Hz, 500 ms) so that IPSCs were not significantly different from baseline. Reactivation with green light (5 min, 0.5 Hz, 500 ms) decreased IPSCs. Alloswitch-1 by itself and its inactivation and reactivation had no significant effects on EPSCs (*Figure 3m and o*). Accordingly, alloswitch-1 decreased the inhibition/excitation (I/E) ratio significantly; this effect was reversed with blue-violet light for optical inactivation and was reinstated with green light for reactivation (*Figure 3p*). The data suggest that mGlu5 blockade reduces BLA-driven feed-forward inhibition onto CeLC neurons that have been linked mechanistically to pain behaviors, which could explain the hypersensitivity observed with photopharmacological blockade of mGlu5 receptors.

The aim of this study was to determine the region-specific behavioral and electrophysiological effects of mGlu5 receptor blockade with caged and photoswitchable NAMs in neuropathic pain using photopharmacology. While the effect of optical manipulations of photosensitive mGlu5 NAMs in individual brain regions in pain models has been studied before (*Font et al., 2017*; *Notartomaso et al., 2022*; *Gómez-Santacana et al., 2017*), the comparative analysis across different brain regions and their contribution to systemically applied NAMs represents a novel concept.

## Discussion

This photopharmacological study significantly advances our understanding of brain region-specific mGlu5 receptor function in pain modulation. To the best of our knowledge, this is the first study to employ photopharmacological tools to compare and contrast distinct roles of mGlu5 receptors in different regions of the pain matrix.

The net effect of systemic administration of alloswitch-1 was the enhancement of pain thresholds, which is in line with findings obtained with other mGlu5 receptor NAMs in models of neuropathic pain (*Lax et al., 2014*; *Jacob et al., 2009*), indicating that the overall effect of the drug across its target regions within the pain neuraxis is the induction of analgesia. However, the combined use of mGlu5 NAMs JF-NP-26 (which is activated by blue-violet light) and alloswitch-1 (which is inactivated by blue-violet light and re-activated by green light) showed for the first time that responses to mGlu5 receptor blockade are not homogenous in different regions of the pain matrix. Increases in pain thresholds after local activation of JF-NP-26 in a specific brain region indicate that mGlu5 receptor blockade in that region is sufficient for the induction of analgesia. In contrast, loss of analgesia after local inactivation of allowitch-1 suggests that mGlu5 receptor blockade in that region is necessary for the overall behavioral effect of a systemic mGlu5 receptor NAM. This innovative approach allowed us to uncover the heterogeneity and complexity of mGlu5 receptor blockade in pain processing, providing valuable insights into the underlying mechanisms of neuropathic pain. By using precise spatial control of mGlu5 receptor function, we were able to dissect the contributions of specific brain regions to the overall analgesic response.

We found that mGlu5 receptor blockade in the contralateral ventrobasal thalamus and infralimbic and prelimbic subregions of the medial prefrontal cortex (mPFC) was sufficient for the induction of analgesia and was also required for the analgesic activity of a systemically administered mGlu5

receptor NAM (*Figure 2*). Blockade of mGlu5 receptors in these regions resulted in increased pain thresholds, indicating their crucial involvement in the analgesic response. Conversely, selective optical inactivation of systemic alloswitch-1 in the infralimbic and prelimbic cortices and thalamus blocked the analgesic effect. This observation is particularly remarkable, considering that alloswitch-1 remained active and blocked mGlu5 receptors in all other regions of the pain system. These data suggest that regulation of pain thresholds by mGlu5 receptors in the ventrobasal thalamus and the two more ventral mPFC subregions is not redundant, but rather complementary, and that a connection between these three regions is a main target for systemic mGlu5 NAMs in the induction of analgesia in models of neuropathic pain. Nociceptive sensitization has been consistently associated with hypoactivity of pyramidal neurons in the prelimbic cortex, and activation of pyramidal neurons causes analgesia in models of neuropathic pain (*Chung et al., 2018*). A simplistic hypothesis is that alloswitch-1 or light-activated JF-NP-26 caused analgesia by antagonizing mGlu5 receptors expressed by inhibitory inter-neurons, or by excitatory neurons that preferentially activate inhibitory interneurons, within the local circuit (*Chung et al., 2018*). In contrast, light-induced inactivation of allowitch-1 in the contralateral anterior cingulate cortex did not abrogate the analgesic activity of the drug. While mGlu5 receptor blockade in the anterior cingulate cortex was not necessary for the overall analgesic effect of systemi-cally administered mGlu5 NAM, antinociceptive effects of mGlu5 receptor blockade with JF-NP-26 in that region suggest that it plays a significant role in modulating pain behaviors.

There are contrasting reports on the role of mPFC mGlu5 receptors in pain modulation (*Chung et al., 2018*; *Kiritoshi et al., 2016*; *Ji and Neugebauer, 2014*; *David-Pereira et al., 2016*; *Palazzo et al., 2014*; *Giordano et al., 2012*). For example, the application of mGlu5 receptor antagonists in the ipsilateral prelimbic and infralimbic cortices enhanced neuropathic pain (spared nerve injury) (*Giordano et al., 2012*), whereas ipsilateral infralimbic application reduced arthritis-induced pain (*David-Pereira et al., 2016*) and bilateral application in prelimbic cortex caused antinociception in the spinal nerve ligation model of neuropathic pain (*Chung et al., 2017*). Our photopharmacological tools allowed us to establish that pharmacological blockade of mGlu5 receptors in the contralat-eral mPFC caused antinociception in the CCI model of neuropathic pain, and that prelimbic and infralimbic regions, but not anterior cingulate cortex, play a critical role in mediating systemic mGlu5 NAM antinociception. Our electrophysiological in vivo recordings provide evidence for the control of pro-nociceptive ON-cells in the RVM by prelimbic output (*Figure 3—figure supplement 1*) that is decreased by mGlu5 receptor inhibition (reinstating the local effects of alloswitch-1 in the prelimbic cortex) in neuropathic pain. Accordingly, our ex vivo brain slice physiology experiments found increased excitatory drive onto prelimbic pyramidal cells (*Figure 3*). Interestingly, blockade of mGlu5 in the thalamus also inhibited pain behaviors (*Figure 2*), possibly through projections to cortical regions such as somatosensory and insular cortices, with the latter projecting strongly to the prelimbic cortex (*Dixsaut and Gräff, 2022*). Previous studies showed increased mGlu5 receptor expression in the mPFC ipsilateral, but not contralateral, to peripheral nerve injury (*Chung et al., 2017*). We used an in vivo method to measure the primary signal transduction mechanism of mGlu5 receptors, i.e., poly-phosphoinositide (PI) hydrolysis. The method was based on a pre-treatment with lithium ions (to block the conversion of inositol phosphate, InsP, into free inositol) followed by systemic activation of mGlu5 receptors with a selective PAM (VU0360172) and measurements of InsP accumulation. We found that mGlu5 receptor-mediated PI hydrolysis was significantly amplified in all subregions of the contralateral mPFC and in the contralateral amygdala after induction of neuropathic pain whereas mGlu5 receptor protein levels were significantly increased only in the contralateral infralimbic cortex of neuropathic mice (*Figure 1*; *Figure 1—figure supplement 1*). This suggests that, at least in the anterior cingulate cortex, prelimbic cortex, and basolateral amygdala, mGlu5 receptors become hyperactive after the induction of pain. It remains to be determined if this is mediated by the enhanced coupling of mGlu5 receptors to $G_{q/11}$ proteins, increased expression of phospholipase-C$\beta$, or other mechanisms. Interest-ingly, mGlu5 receptor signaling was down-regulated in the thalamus of neuropathic mice, but mGlu5 blockade in the thalamus still had antinociceptive effects (see below). Down-regulation of mGlu5 receptor signaling in the thalamus might represent a compensatory mechanism aimed at mitigating pain in neuropathic mice.

By elucidating the regional specificity of mGlu5 receptor modulation, this study expands our understanding of the neural circuitry involved in pain processing (*Figure 4*). Our findings have impli-cations for the development of targeted therapies for pain management. Selective modulation of

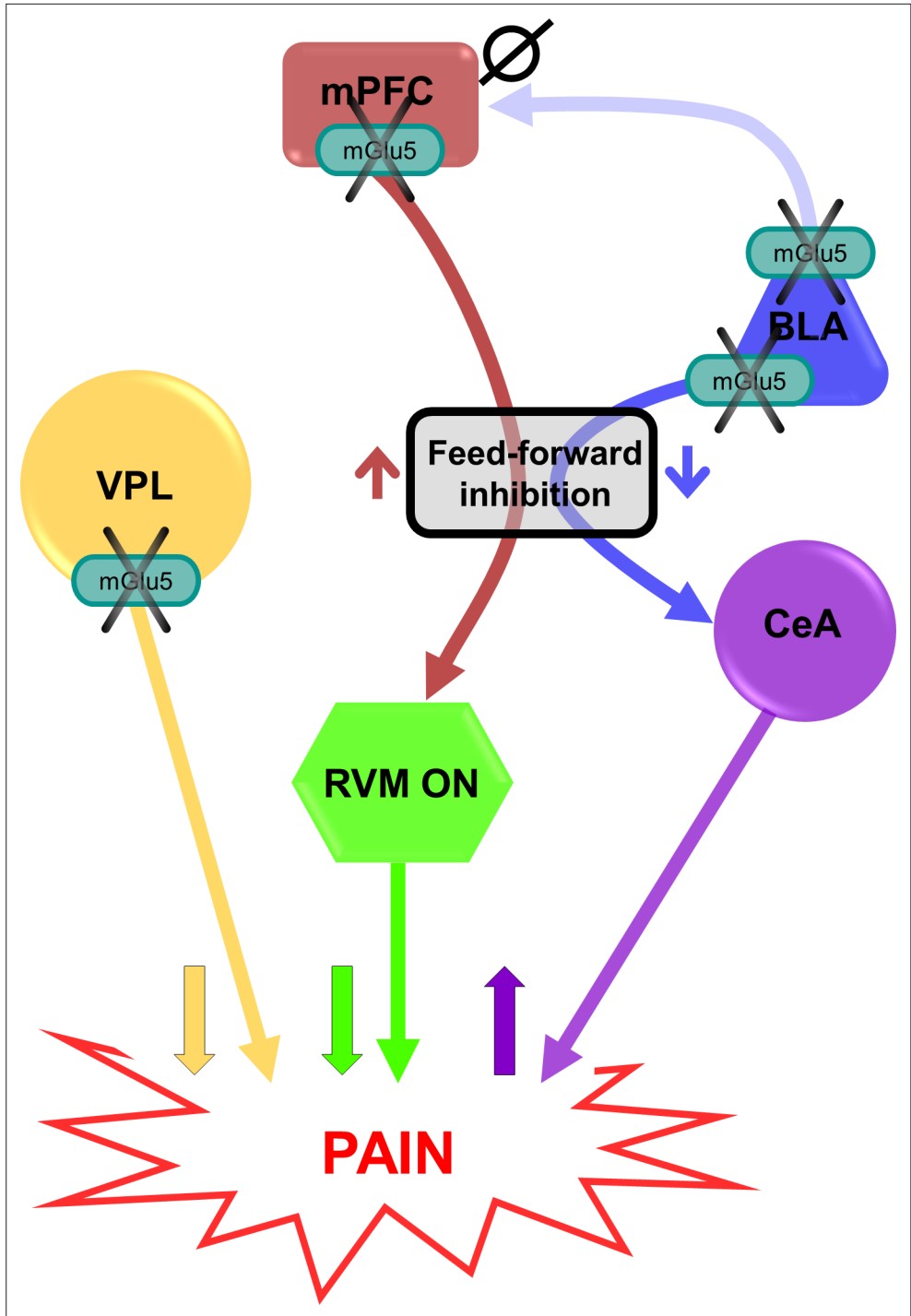

**Figure 4.** Putative circuitry of brain region-specific functions of mGlu5 receptors in pain modulation. Hypothesized neural circuitry based on in vivo and ex vivo electrophysiology and behavioral data. mGlu5 receptor blockade decreases feedforward inhibition from BLA to CeA to increase amygdala output (*Figure 3i–p*) and facilitate pain behaviors (*Figure 2g and m*, and *Figure 2—figure supplement 3*), but has no effect on mPFC activity (*Figure 3—figure supplement 1d-f*). mGlu5 receptor blockade in the mPFC increases pyramidal cell output (*Figure 3a–h*) to decrease the activity of pronociceptive RVM ON-cells via feedforward inhibition, resulting in decreased pain behavior (*Figure 2e and k*). mGlu5 receptor blockade in the VPL resulted in antinociceptive effects (*Figure 2h and n*). mPFC = medial prefrontal cortex; VPL = ventral posterolateral nucleus of the thalamus; BLA = basolateral amygdala; CeA = central nucleus of amygdala; RVM = rostral ventromedial medulla; ↑ facilitation; ↓ inhibition; Ø no effect.

mGlu5 receptor activity in the infralimbic and prelimbic cortex presents an intriguing opportunity to harness their specific analgesic effects while minimizing potential side effects associated with global mGlu5 receptor blockade. The differential roles of mPFC regions in pain modulation highlight their unique contributions to the complex pain circuitry. Targeting the infralimbic and prelimbic cortices allows for a more precise and localized approach, potentially reducing the risk of off-target effects and enhancing the therapeutic efficacy of mGlu5 receptor-based interventions. This selective modulation strategy holds promise for the development of targeted and personalized treatments for neuropathic pain, offering a potential avenue for improved pain management with reduced adverse effects. Future research aimed at elucidating the specific mechanisms underlying the analgesic effects of mGlu5 receptor modulation in these cortices will pave the way for the development of novel therapeutic interventions and optimization of pain management strategies.

A somewhat surprising finding of our study was the unique role of the amygdala (BLA) in mGlu5 function in neuropathic pain. Photopharmacology revealed for the first time that the BLA is a voice out of the chorus in the modulation of pain by mGlu5 receptors in neuropathic mice. Light-induced activation of JF-NP-26 in the right BLA reduced pain thresholds in neuropathic mice, indicating pronociceptive effects, whereas light-induced inactivation of alloswitch-1 in the BLA further enhanced analgesia (*Figure 2*). We also found that JF-NP-26 activation in the BLA decreased risk-taking behavior in neuropathic mice, consistent with increased anxiety (*Figure 4*). This suggests that mGlu5 receptors in BLA neurons serve antinociceptive functions in neuropathic pain, and this may limit the overall effect of mGlu5 NAMs in the treatment of pain. One possible explanation we considered is that mGlu5 receptors in the BLA modulate connections with the prelimbic and infralimbic cortices. These connections involve direct excitation as well as feedforward inhibition, and previous studies showed that electrical stimulation of BLA neurons enhanced inhibition of prelimbic neurons (*Guida et al., 2017*; *Ishikawa and Nakamura, 2003*) and decreased excitation/inhibition balance in pain resulted in impaired mPFC output and behavioral control (*Kiritoshi et al., 2016*; *Cheriyan and Sheets, 2020*). Therefore, mGlu5 NAM could be pronociceptive by decreasing the excitation/inhibition balance of BLA inputs to prelimbic regions. Results from our in vivo electrophysiological experiments, however, do not support this hypothesis (*Figure 3—figure supplement 1*). Instead, our ex vivo brain slice physiology studies found that mGlu5 NAM decreased feedforward inhibition of neurons in the CeA, an important output region for behavioral modulation (*Neugebauer, 2020*; *Figure 3*). Impaired feedforward inhibition of CeA output neurons drives pain behaviors (*Ren et al., 2013*). The role of mGlu5 function in the amygdala appears to be complex. Our data suggest pronociceptive effects of mGlu5 NAM in BLA through connections with CeA, whereas previous studies reported antinociceptive effects of mGlu5 receptor blockade with MPEP in the CeA in inflammatory pain models (*Han and Neugebauer, 2005*; *Kolber et al., 2010*). Administration of alloswitch-1 into CeA also decreased hypersensitivity in an inflammatory pain model (*Ricart-Ortega et al., 2020*). Interestingly, a mGlu5 antagonist (MTEP) inhibited both excitatory and inhibitory transmission in CeA neurons under normal conditions and in an arthritis pain model (*Ren and Neugebauer, 2010*). Thus, differential mGlu5 function in BLA and CeA may serve to shed light onto the intricate amygdala circuitry involved in pain processing and pain modulation.

In summary, the application of photopharmacology to the analysis of mGlu5 function in pain discovered novel region-specific differential actions that need to be considered when targeting these receptors as novel and improved therapeutic targets, and this may have implications for other transmitter and modulator systems.

## Materials and methods

### Drugs

JF-NP-26 (10 mg/kg, i.p. in 6% DMSO, 6% Tween 80 in saline) and alloswitch-1 (10 mg/kg, i.p. in 20% DMSO, 20% Tween 80 in saline) were kindly provided by Professor Llebaria (IQAC-CSIC, Barcelona, Spain). Lithium chloride was purchased from Sigma-Aldrich (Milan, Italy). VU0360172 [N-cyclobutyl-6-(2-(3-fluorophenyl)ethynyl) pyridine-3-carboxamide] was purchased from Tocris Bioscience (Bristol, United Kingdom).

### Animals

All experiments were carried out according to the European (2010/63/EU) and Italian (D. Lgs. 26/2014) guidelines for animal care. The Italian Ministry of Health (n° 804/2018-PR) approved the experimental

protocol. Experiments carried out at Texas Tech University Health Sciences Center (TTUHSC) were approved by the Institutional Animal Care and Use Committee (IACUC) at TTUHSC (14006) and conform to the guidelines of the International Association for the Study of Pain (IASP) and National Institutes of Health (NIH). Every effort was made to minimize animal suffering and the number of animals used. Adult male C57BL/6 J mice (20–25 g, b.w.) were purchased from Charles River (Calco, Italy) for the behavioral assays and in vivo electrophysiological recordings, while wild-type C57BL/6 J mice (original breeding pairs from Jackson lab) were used for brain slice electrophysiology experiments. All animals were housed in groups of four in standard cages with ad-libitum access to food and water and maintained under 12 hr dark/light cycle (starting at 7:30 AM), 22 °C temperature, and 66% humidity (standard conditions). Sample size was based on our previous studies and experience with similar behavioral, electrophysiological, and molecular assays. Animals were randomly assigned to different experimental groups and a researcher blinded to drug treatments carried out all animal experimentation. All the experiments were conducted by at least two investigators. Some of the electrophysiological experiments represent replicates.

## CCI model of neuropathic pain and assessment of mechanical allodynia

Induction of the sciatic nerve CCI model (*Bennett and Xie, 1988*) and assessment of mechanical hypersensitivity are described in more detail in Bio-protocol (*Notartomaso et al., 2018*). In brief, CCI induction was carried out under isoflurane anesthesia (5% for induction and 2% for maintenance), using a modified version of a previously described method (*Bennett and Xie, 1988*). The biceps femoris and the gluteus superficialis were separated by blunt dissection, and the left sciatic nerve was exposed. CCI was produced by tying two ligatures (6–0 silk, Ethicon, LLC, San Lorenzo, PR, USA) around the sciatic nerve. The ligature was tied loosely around the nerve, until it elicited a brief twitch in the respective hind limb. Over-tightening of the ligation was avoided to preserve epineural circulation. The incision was cleaned and the skin was closed with 2–3 ligatures of 5–0 dermalon. In sham-operated mice, the left sciatic nerve was exposed without ligature. Mechanical allodynia was assessed 16 days after surgery by measuring the hind paw withdrawal response to von Frey filament stimulation. Mice were placed in a dark box (20 × 20 × 40 cm) with a wire grid bottom through which the von Frey filaments (North Coast Medical, Inc, San Jose, CA, USA; bending force ranging from 0.008 to 3.5 g) were applied by using a modified version of the up-down paradigm previously described (*Chaplan et al., 1994*). Each filament was applied five times (3 min intervals) perpendicularly to the plantar surface of the hind paw until it bent. The filament that evoked at least three paw withdrawals was assigned as the pain threshold in grams.

## Stereotaxic implantation of optic fibers in different brain regions

Mice were anesthetized with isoflurane (5% for induction and 2% for maintenance) and implanted with monolateral LED optic fibers (MFC_400/430–0.57_mm_SMR_C45 Doric Lenses Inc, Quebec, Canada) using dental cement and surgical screws (Agnthos, Lidingö, Sweden) in a stereotaxic frame (Stoelting Co., Wood Dale, IL, USA). The following brain regions in the right hemisphere were targeted (*Paxinos and Franklin, 2008*): prelimbic cortex (1.50 mm anterior to the bregma, 0.3 mm lateral to midline, 2.7 mm ventral from surface of the skull), infralimbic cortex (1.70 mm anterior, 0.3 mm, 3 mm), anterior cingulate cortex (0.5 mm anterior, 0.2 mm, 2 mm), basolateral amygdala (1.34 mm posterior, 2.9 mm, 4.7), ventral posterolateral thalamus (1.8 mm posterior, 1.5 mm, 3.5 mm). The length (mm) of the optic fibers was selected according to the targeted brain area. Previous studies from our group showed that stereotaxic insertion of optical fibers into the infralimbic cortex leaves cortical connectivity intact (*Ji and Neugebauer, 2012*).

## Optical manipulation of drugs in different brain regions

Mice were treated with JF-NP-26, (10 mg/kg, i.p. in 6% DMSO, 6% Tween 80 in saline) or alloswitch-1 (10 mg/kg, i.p. in 20% DMSO, 20% Tween 80 in saline). Individual brain areas (infralimbic cortex, prelimbic cortex, anterior cingulate cortex, ventrobasal thalamus, and basolateral amygdala contralateral to the side of injury) were illuminated through the implanted optic fiber with conical tip (Doric Lenses Inc) with blue-violet light (405±6.5 nm) to activate JF-NP-26 or inactivate alloswitch-1 and with green light (520±19 nm) to reactivate alloswitch-1 (*Pittolo et al., 2014*) (see *Figure 2c*). The selection of wavelengths was based on our previous work (*Pittolo et al., 2014*; *Ricart-Ortega et al., 2020*).

Pulsed light illumination (or dark control) was for 5 min (0.5 Hz, 500ms), 20 min after drug administration of the caged or photoswitchable compound. The light spread with a conical tip cannula is less than 200 μm according to the manufacturer (Doric Lenses Inc).

## Light-dark box and tail suspension test (TST)

The light-dark box consisted of two compartments: an illuminated compartment and a dark compartment (50 × 20 cm) connected by a shuttle door (12 × 5 cm) in the center of the partition at floor level. The light box was open at the top, painted white, and illuminated by a 60 W light bulb located 30 cm above the apparatus, providing illumination in the range of 380–470 lux measured on the floor of the white compartment. The dark box was painted matte black and had a removable black lid at the top. Each animal was placed at the center of the dark compartment and was allowed to explore freely both compartments for 10 min. Total number of crossings between the two compartments (defined as all four paws being within a given compartment), latency to enter the illuminated compartment, and time spent in the illuminated compartment were measured. Movement of each animal was recorded using a video camera and analyzed by a blinded operator. Neuropathic mice were treated with JF-NP-26 (10 mg/kg, i.p. in 6% DMSO, 6% Tween 80 in saline) with or without violet light delivered into the BLA. In the TST, each mouse was suspended by its tail using adhesive tape and attached to a hook. The duration of immobility during the 5 min test period was measured.

## Western blot analysis

Tissues were dissected out and homogenized at 4 °C in RIPA buffer containing protease inhibitors cocktail (Merck Millipore, Milano, Italy) for 30 min, and an aliquot was used for protein determination. Equal amounts of proteins (20 μg) from supernatants were separated by 8% SDS polyacrilamide gel at 100 V for 1 hr for the detection of mGlu5 receptors, using a mini-gel apparatus (Bio-Rad Mini Protean II cell, Milano, Italy). Proteins were then electroblotted on Immuno PVDF membranes (Bio-Rad) for 7 min using a Trans Blot Turbo System (Bio-Rad). Filters were washed three times and blocked for 1 hr in Tris-Tween buffered saline (TTBS) containing 5% non-fat dry milk. The following primary antibodies were used: rabbit monoclonal anti-mGlu5 receptor antibody (1:200, Abcam, Cambridge, UK, Cat #AB76316, lot. GR45647-18). Filters were washed three times with TTBS buffer and then incubated for 1 hr with secondary peroxidase-coupled anti-rabbit antibody (1:7000, Millipore, Cat #401393–2 ML). Immunostaining was revealed by enhanced chemiluminescence luminosity (Amersham Pharmacia Biotech, Arlington Height, IL). The blots were re-probed with monoclonal anti-β-actin antibody (1:50000, Sigma-Aldrich, Cat # A5441, lot. 116M4801V). Statements of validation and references are on the manufacturers' websites; antibodies were validated in our previous studies, including with knock-out mice (PMID: 30190524). ImageLab 6.1 software was used for the acquisition and analysis of the Western blot image data.

## ELISA

For the assessment of mGlu5 receptor signaling, mice were injected systemically with lithium chloride (10 mM, 100 μl, i.p.) followed, after 1 hr, by i.p. injection of a positive allosteric modulator for mGlu5 (VU0360172, 30 mg/kg i.p.) to activate mGlu5 receptors. Control mice were treated with lithium chloride followed by the vehicle. Mice were sacrificed 60 min after the last injection and the brain regions were microdissected with a Vibratome (Leica Biosystems, Buccinasco, MI, Italy). The selected brain regions were weighed and homogenized by sonication in 10 μl/mg of tissue of Tris-HCl buffer (100 mM; pH 7.5) containing 150 mM NaCl, 5 mM EDTA, 1% Triton X-100, 1% SDS. Homogenates were diluted 1:50 and InsP levels were assessed with the IP-One ELISA kit (Cisbio, Codolet, France) according to the manufacturer's instructions. Microplate manager 6 software (MPM6 Biorad) was used for plate reading.

## Patch-clamp electrophysiology in brain slices

Whole-cell patch-clamp recordings were performed in brain slices containing the right medial prefrontal cortex or amygdala obtained from CCI mice (16 days after surgery) as described previously (*Ji et al., 2010*; *Ren et al., 2013*; *Yakhnitsa et al., 2022*). Brains were quickly removed and immersed in an oxygenated ice-cold sucrose-based physiological solution containing (in mM): 87 NaCl, 75 sucrose, 25 glucose, 5 KCl, 21 MgCl$_2$, 0.5 CaCl$_2$, and 1.25 NaH$_2$PO$_4$. Coronal brain slices of 300 μm thickness

were obtained using a Vibratome (VT1200S, Leica Biosystems VT Series, Nussloch, Germany) and incubated in oxygenated artificial cerebrospinal fluid (ACSF, in mM: 117 NaCl, 4.7 KCl, 1.2 NaH$_2$PO$_4$, 2.5 CaCl$_2$, 1.2 MgCl$_2$, 25 NaHCO$_3$ and 11 glucose) at room temperature (21 °C) for at least 1 hr before patch recordings. A single brain slice was transferred to the recording chamber and submerged in ACSF (31 ± 1°C) superfusing the slice at ~2 ml/min. Only one or two brain slices per animal were used and only one neuron was recorded in each brain slice. Whole-cell patch-clamp recordings were made from visually identified prelimbic or central amygdala (laterocapsular division) neurons using DIC-IR videomicroscopy as described previously (*Ji et al., 2010*; *Ren et al., 2013*). Recording electrodes (tip resistance 5–8 MΩ) were made from borosilicate glass and filled with intracellular solution containing (in mM): 122 K-gluconate, 5 NaCl, 0.3 CaCl$_2$, 2 MgCl$_2$, 1 EGTA, 10 HEPES, 5 Na$_2$-ATP, and 0.4 Na$_3$-GTP; pH was adjusted to 7.2–7.3 with KOH and osmolarity to 280 mOsm/kg with sucrose. In some experiments, 0.2% biocytin was included in the intracellular solution on the day of the experiment. Data acquisition and analysis were done using a dual 4-pole Bessel filter (Warner Instr., Hamden, CT), low-noise Digidata 1322 or 1550B interface (Axon Instr., Molecular Devices, Sunnyvale, CA), Axoclamp-2B or MulitClamp700B amplifier (Axon Instr., Molecular Devices, Sunnyvale, CA), and pClamp9 or 11 software (Axon Instr.). If series resistance (monitored with pClamp9 or 11 software) changed by more than 20%, the neuron was discarded. To characterize the properties of recorded neurons, depolarizing current pulses (500 ms, 25 pA step) were applied in current clamp mode. Neurons were voltage-clamp at –70 mV or 0 mV to study excitatory and inhibitory postsynaptic currents (EPSCs and IPSCs) evoked by the electrical stimulation of presumed afferents from basolateral amygdala (BLA) using a concentric bipolar stimulating electrode (David Kopf Instruments). Monosynaptic EPSCs and glutamate-driven IPSCs were completely blocked by the application of a non-NMDA receptor antagonist (6-cyano-7-nitroquinoxaline-2,3-dione, CNQX, 20 µM). IPSCs were blocked by bicuculline (10 µM). Alloswitch-1 was applied to the brain slice by gravity-driven superfusion in ACSF (final concentration 100 nM *Pittolo et al., 2014*, 2 ml/min), and electrophysiological measurements were made 10–15 min during drug application. Illumination of the target region in the slice was done through the objective of the microscope (BX51, Olympus, Waltham, MA) with an integrated LED stimulation system (X-Cite Xylis). Photoisomerization of alloswitch-1 was achieved by illumination with blue-violet light (375±14 nm) generated by a broad-spectrum LED illumination system by passing through an AT375/28 x filter (Chroma Technology Corp., Bellows Falls, VT) for 5 min (0.5 Hz, 500 ms), and was reversed by illumination with green light (545±15 nm) an AT545/30 x filter (Chroma Technology Corp., Bellows Falls, VT) for 5 min (0.5 Hz, 500 ms). The electrophysiological outcome measures were assessed after 5 min and during (blue-violet or green) light application.

## In vivo electrophysiology

Single-unit electrophysiological recordings were performed in the prelimbic cortex or RVM in CCI mice 16 days after injury. An LED optic fiber was implanted into the BLA when recording from the prelimbic area, or into the prelimbic cortex when recording from RVM. Mice were anaesthetized with an intra-peritoneal injection of ketamine (100 mg/kg) and xylazine (10 mg/kg) and anesthesia was maintained with a constant continuous infusion of propofol (5–10 mg/kg/h, i.v.) throughout the experiment. A small craniotomy allowed the stereotaxic insertion of a glass-insulated tungsten filament recording electrode (3–5 MΩ) (Frederick Haer & Co., Bowdoin, ME, USA) that was lowered into the prelimbic cortex (1.50 mm anterior to the bregma, 0.3 mm lateral to midline, 1.5–2.7 mm ventral from surface of the skull) or RVM (6.48 mm posterior to bregma, 0.3–0.5 mm lateral, and 4.5–6 mm ventral) as previously described (*Guida et al., 2018*; *Boccella et al., 2019*). Spike waveforms were displayed on an oscilloscope to ensure that the unit under study was unambiguously discriminated throughout the recording period. Signals were processed using an interface (CED 1401; Cambridge Electronic Design Ltd., Cambridge, United Kingdom) connected to a Pentium III PC. Spike2 software (CED, v.4) was used to create peristimulus rate histograms on-line and to store and analyze digital records of single-unit activity off-line. In the RVM, we classified ON- and OFF-cells based on their firing patterns relative to nocifensive withdrawal responses (*Fields and Heinricher, 1985*). As ON-cells with a high basal firing rate can be easily misclassified as NEUTRAL-cells (*Barbaro et al., 1986*), potential NEUTRAL-cells with continuous spontaneous activity were verified by giving a brief bolus of anesthetic to the point that the withdrawal reflex was abolished. Indeed, the firing of spontaneously active ON-cells slows or stops with this manipulation, which unmasks reflex-related responses. Recordings were performed only from

ON-cells, which have response characteristics consistent with a role in pain facilitation and accelerate firing immediately before the nocifensive reflex (*Chen and Heinricher, 2022*).

Specifically, cells were identified by a burst of activity in response to a brief (3 s) noxious mechanical stimulus (von Frey filament with bending force of 97.8 mN) applied to the hind paw contralateral to the RVM. RVM ON-cell burst activity (Hz) evoked by peripheral noxious stimulation was calculated as number of spikes in the 10 s interval starting from the beginning of the increase in firing frequency (which was at least double its spontaneous activity). In the prelimbic cortex, regular-spiking pyramidal cells were studied. Background activity (Hz) refers to neuronal activity in the absence of intentional stimulation. Neuronal activity (frequency of excitation of RVM neurons or background activity of prelimbic neurons) was measured every 5 min before and after systemic administration of alloswitch-1 and before and after optical manipulation, and was calculated as mean ± SEM.

## Statistical analysis and data presentation

Statistical analysis was performed with GraphPad Prism version 9.5.1 for Windows (GraphPad Software, San Diego, California USA). Standard two-tailed unpaired or paired student t-test and one or two-way ANOVA followed by Bonferroni or Dunnett's posthoc tests were used as appropriate. A value of $P<0.05$ was considered statistically significant. All data are presented in the Figures. No data from experiments presented in this study were excluded, and all data represent biological replicates.

## Acknowledgements

This work was supported by Ricerca Finalizzata of Italian Ministry of Health—Young Researcher project (GR-2016–02362046) to SN, AL received funding from Ministerio de Ciencia e Innovación, Agencia Estatal de Investigación 10.13039/501100011033 and ERDF A way of making Europe (Projects I+D + i CTQ2017-89222-R and PID2020-120499RB-I00), CSIC (201980E011) and Departament de Recerca i Universitats, Generalitat de Catalunya (2017SGR1604 and 2021SGR00508). VN received funding from National Institutes of Health (NIH) grant R01 NS038261.

## Additional information

### Funding

| Funder | Grant reference number | Author |
|---|---|---|
| Ministero della Salute | GR-2016-02362046 | Serena Notartomaso |
| Ministerio de Ciencia e Innovación | 10.13039/501100011033 | Amadeu Llebaria |
| ERDF A way of making Europe | Projects I+D+i CTQ2017-89222-R and PID2020-120499RB-I00 | Amadeu Llebaria |
| CSIC | 201980E011 | Amadeu Llebaria |
| Departament de Recerca i Universitats Generalitat de Catalunya | 2017SGR1604 and 2021SGR00508 | Amadeu Llebaria |
| National Institutes of Health | R01 NS038261 | Volker Neugebauer |

The funders had no role in study design, data collection and interpretation, or the decision to submit the work for publication.

### Author contributions

Serena Notartomaso, Conceptualization, Formal analysis, Investigation, Visualization, Methodology, Writing - original draft; Nico Antenucci, Formal analysis, Investigation, Visualization, Methodology, Writing - original draft, Writing - review and editing; Mariacristina Mazzitelli, Formal analysis, Investigation, Visualization, Methodology, Writing - original draft; Xavier Rovira, Resources, Methodology; Serena Boccella, Formal analysis, Investigation, Visualization, Writing - original draft; Flavia Ricciardi,

Tiziana Imbriglio, Milena Cannella, Formal analysis, Investigation; Francesca Liberatore, Formal analysis, Investigation, Visualization; Xavier Gomez-Santacana, Resources; Charleine Zussy, Livio Luongo, Giuseppe Battaglia, Conceptualization, Supervision, Methodology; Sabatino Maione, Conceptualization, Supervision, Project administration; Cyril Goudet, Conceptualization, Supervision, Methodology, Writing - review and editing; Amadeu Llebaria, Conceptualization, Resources, Supervision, Methodology, Writing - review and editing; Ferdinando Nicoletti, Conceptualization, Resources, Supervision, Funding acquisition, Project administration, Writing - review and editing; Volker Neugebauer, Conceptualization, Resources, Supervision, Funding acquisition, Investigation, Methodology, Project administration, Writing - review and editing

#### Author ORCIDs
Serena Notartomaso ⓘ https://orcid.org/0000-0003-4374-9233
Mariacristina Mazzitelli ⓘ http://orcid.org/0000-0001-5122-6649
Xavier Rovira ⓘ https://orcid.org/0000-0002-9764-9927
Cyril Goudet ⓘ https://orcid.org/0000-0002-8255-3535
Giuseppe Battaglia ⓘ https://orcid.org/0000-0001-7571-3417
Amadeu Llebaria ⓘ https://orcid.org/0000-0002-8200-4827
Ferdinando Nicoletti ⓘ https://orcid.org/0000-0003-0917-443X
Volker Neugebauer ⓘ https://orcid.org/0000-0001-6158-8635

#### Ethics
All experiments were carried out according to the European (2010/63/EU) and Italian (D. Lgs. 26/2014) guidelines of animal care. The Italian Ministry of Health (n° 804/2018-PR) approved the experimental protocol. Experiments carried out at Texas Tech University Health Sciences Center (TTUHSC) were approved by the Institutional Animal Care and Use Committee (IACUC) at TTUHSC (14006) and conform to the guidelines of the International Association for the Study of Pain (IASP) and National Institutes of Health (NIH).

#### Decision letter and Author response
Decision letter https://doi.org/10.7554/eLife.94931.sa1
Author response https://doi.org/10.7554/eLife.94931.sa2

## Additional files

#### Supplementary files
• MDAR checklist

• Supplementary file 1. Effect size of optical modulation in prelimbic cortex on RVM activity, according to Cohen's d calculation from t-tests.

#### Data availability
All data needed to evaluate the conclusions in the paper are presented in the paper and/or the Supplementary Materials, and have been provided as excel files.

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
