## [Editor Report]

In this convincing study, the authors have used light-sensitive mGlu5 negative allosteric modulators to determine the role of these receptors in a chronic pain model. These findings are have important implications for the pain field, and will generally valuable to neuroscientists interested in signaling through mGluR5 receptors.

---

## [Decision Letter]

**Decision letter after peer review:**

Thank you for submitting your article "A 'double-edged' role for type-5 metabotropic glutamate receptors in pain disclosed by light-sensitive drugs" for consideration by *eLife*. Your article has been reviewed by 3 peer reviewers, and the evaluation has been overseen by a Reviewing Editor and Sacha Nelson as the Senior Editor.

The reviewers have discussed their reviews with one another, and all agree the work has implications beyond the pain field and has the potential to be broadly important. However, they felt the main claims are only partially supported by the data and therefore incomplete. The three main points that should be addressed are see below. We are also attaching the full reviews below for details that should help you prepare your revision.

Essential revisions (for the authors):

(1) Conduct behavioral tests that measure the "affective" components of pain. The behaviors examined need to be expanded beyond reflexive assays, for example, examine place preference/aversion.

(2) Provide better validation of the light-activated agents, ideally in vivo.

(3) Add missing controls, increase sample sizes, and perform the correct statistical tests. Furthermore, you should better explain/discuss the effect sizes and justify why only male mice are studied (or, if possible, add female mice).

*Reviewer #2 (Recommendations for the authors):*

In this study, Notartomaso et al. used optical activation of systemic JF-NP-26, a caged, baseline inactive, negative allosteric modulator (NAM) of mGlu5 receptors, in cingulate, prelimbic and infralimbic cortices, thalamus, and BLA to investigate the roles of these receptors in various brain regions in pain processing. They found that alloswitch-1, an intrinsically active mGlu5 receptor NAM, caused analgesia, but this analgesic effect was reversed by light-induced drug inactivation in the prelimbic and infralimbic cortices, and thalamus. In contrast, these pharmacological effects were reversed in the BLA. They further found that alloswitch-1 increased excitatory synaptic responses in prelimbic pyramidal neurons evoked by stimulation of BLA input, and decreased feedforward inhibition of amygdala output neurons by BLA. They thus concluded that mGlu5 receptors had differential effects in distinct brain regions. mGlu5 receptors are important receptors in pain processing, and their regional specificity has not been studied in detail. Further, this is an interesting study regarding the use of optical activation of pro-drugs, and the findings are timely.

I have the following comments that the author may find helpful.

– It may be helpful to elucidate the range of light activation, as the subregions of mPFC are rather close to each other in mice. Likewise, it may be helpful to show histology for fiber locations.

– The role of ACC in pain processing is typically affective and thus is tested by conditioned place aversion/preference assays, and so the authors may consider the application of these assays to further validate their claims.

– The acronym π was not spelled out.

*Reviewer #3 (Recommendations for the authors):*

In this manuscript, Notartomaso, Antenucci et al. use two different light-sensitive metabotropic glutamate receptor negative allosteric modulators (NAMs) to determine how mGlu5 receptor signaling in distinct brain regions of mice influences mechanical sensitivity in chronic constriction injury (CCI) model of neuropathic pain. This is an extension of their previous work using photocaged mGlu5 negative allosteric modulators and incorporates a systemically active NAM that can be locally photoswitched off and on with violet and green light, respectively. The authors found that NAM signaling in the thalamus and prefrontal cortical regions consistently reduced mechanical hypersensitivity. However, they observed divergent effects on these measures in the basolateral amygdala. The authors attempted to solve the discrepancy in behavioral measurements between mGlu5 signaling in the basolateral amygdala by determining how NAMs modulate synaptic transmission or in vivo firing and found that these effects were projection-dependent. Overall this paper presents some interesting observations, but the lack of suitable control groups, limited behavior tests, and the broad profiling approach lacks some cohesion and makes it challenging to determine where and how mGlu5 receptors are exerting these effects.

The behavior and slice physiology experiments are both lacking control groups. Given the differences in basal and mGlu5 PAM stimulated InsP levels in figure 1, including sham groups for the behavioral testing could provide further insight into how mGlu5 NAMs influence nociception. At a minimum, at least one vehicle-treated and photostimulated group should be included to rule out potential off-target effects on the contralateral brain region to the nerve injury. This could also help address whether mGlu5 NAMs are directly analgesic or anti-nociceptive.

Does photostimulation or systemic NAM injection impact other behaviors outside of mechanical withdrawal thresholds? This is important for the interpretation of these effects as pro/anti-nociceptive and not due to impacting vigilance, fear, anxiety, etc. which can affect these sensory measurements.

I appreciate the effort to link the differences in behavioral responses with NAM activation in the amygdala using slice physiology. In the proposed models, mGlu5 is proposed to exist on some upstream projections. However, no direct effects were observed from these proposed inhibitory (Figure 3a-h) or excitatory (Figure i-p) populations, arguing against this. In the mPFC model in Figure 3, the authors are measuring a monosynaptic EPSC (presumably) from the BLA. If this model is correct, wouldn't this excitatory input also be enhanced onto the proposed mGlu5+ inhibitory neurons leading to an increase in polysynaptic IPSC amplitudes? Similarly, if the NAM is reducing mGlu5 tone on the excitatory BLA projections to feed-forward inhibitory circuits in the CeA, wouldn't a reduction in monosynaptic EPSC amplitudes also be observed?

Related to the point above, does Alloswitch-1 modulate membrane properties of the recorded output neurons and might there also be a direct effect on these microcircuits?

What light intensities were used, and does violet illumination affect cellular health?

---

## [Author Response]

Essential revisions (for the authors):(1) Conduct behavioral tests that measure the "affective" components of pain. The behaviors examined need to be expanded beyond reflexive assays, for example, examine place preference/aversion.

We performed additional experiments to evaluate depression‐like and anxiety‐like behaviors (Figure S4). Details can be found in our response to reviewer #1.

(2) Provide better validation of the light-activated agents, ideally in vivo.

All the features of light activation (such as duration, intensity, and wavelength range) were selected based on our previous experience. Additional details can be found in our response to reviewer #2 and reviewer #3.

(3) Add missing controls, increase sample sizes, and perform the correct statistical tests. Furthermore, you should better explain/discuss the effect sizes and justify why only male mice are studied (or, if possible, add female mice).

We added all data with inactive and light‐activated JF‐NP‐26 in sham‐operated mice in the new supplementary Figure S3. We performed new experiments to increase the sample size in the π experiments in the infralimbic and prelimbic cortices where the n was low. Additional details can be found in our response to reviewer #1. Sex differences were not the goal of this study. Male mice were used for this photopharmacology study of region‐specific drug effects to avoid complex effects of the estrous cycle, especially because of evidence for mGlu5 receptor interactions with estrogen receptors and complex influences of estrogens and progesterone. Additional details can be found in our response to reviewer #1.

Reviewer #2 (Recommendations for the authors):In this study, Notartomaso et al. used optical activation of systemic JF-NP-26, a caged, baseline inactive, negative allosteric modulator (NAM) of mGlu5 receptors, in cingulate, prelimbic and infralimbic cortices, thalamus, and BLA to investigate the roles of these receptors in various brain regions in pain processing. They found that alloswitch-1, an intrinsically active mGlu5 receptor NAM, caused analgesia, but this analgesic effect was reversed by light-induced drug inactivation in the prelimbic and infralimbic cortices, and thalamus. In contrast, these pharmacological effects were reversed in the BLA. They further found that alloswitch-1 increased excitatory synaptic responses in prelimbic pyramidal neurons evoked by stimulation of BLA input, and decreased feedforward inhibition of amygdala output neurons by BLA. They thus concluded that mGlu5 receptors had differential effects in distinct brain regions. mGlu5 receptors are important receptors in pain processing, and their regional specificity has not been studied in detail. Further, this is an interesting study regarding the use of optical activation of pro-drugs, and the findings are timely.I have the following comments that the author may find helpful.– It may be helpful to elucidate the range of light activation, as the subregions of mPFC are rather close to each other in mice. Likewise, it may be helpful to show histology for fiber locations.

The range of light activation was selected on the basis of our previous experience. For example, transition of alloswtich‐1 from trans into cis (inactivation of alloswitch‐1) is optimal at wavelengths ranging from 385‐390 to 400 (Pittolo et al. Nat. Chem. Biol. 2014, Ricart‐Ortega et al., ACS Pharmacol. Transl. Sci. 2020) (our selected wavelength was 405 ± 6.5). For the cis‐trans isomerization we used wavelengths of 520 + 19 nm, because the optimal wavelength giving a higher proportion of trans isomer in the photostationary state is between 500 and 550 nm (Ricart‐Ortega et al., ACS Pharmacol. Transl. Sci. 2020). Please, see also response to reviewer 3. Information has been added on page 11.

With regard to the potential spread of light (“range”), the light spread with a conical tip cannula is less than 200 micrometers (see details at the manufacturer’s website https://doriclenses.com/downloads/ApplicationNotes/ApplicationNote_Illumination_FiberTip_V1.0.0.pdf). This is now stated on page 11.

We used well established coordinates for stereotaxic injections and ensured consistency by using mice of similar age and weight. As mentioned in Methods, cannulas were custom‐designed with specific lengths appropriate for each targeted region to ensure accurate placement. We checked the correct position of the fiber in pilot experiments to ensure the precision of the area of illumination. We have decades of experience with stereotaxic implantations and injections into different brain regions, but admittedly we did not perform a histological analysis in mice used for photopharmacological experiments, partially because tissues were collected for molecular analyses.

– The role of ACC in pain processing is typically affective and thus is tested by conditioned place aversion/preference assays, and so the authors may consider the application of these assays to further validate their claims.

Please, see our response to Reviewer #1. Just as a note on the side, the ACC has also been linked to the modulation of sensory aspects of pain (thresholds); see for example work from Dr. Min Zhuo.

– The acronym π was not spelled out.

Done (page 3, line 9). “polyphosphoinositide”

Reviewer #3 (Recommendations for the authors):In this manuscript, Notartomaso, Antenucci et al. use two different light-sensitive metabotropic glutamate receptor negative allosteric modulators (NAMs) to determine how mGlu5 receptor signaling in distinct brain regions of mice influences mechanical sensitivity in chronic constriction injury (CCI) model of neuropathic pain. This is an extension of their previous work using photocaged mGlu5 negative allosteric modulators and incorporates a systemically active NAM that can be locally photoswitched off and on with violet and green light, respectively. The authors found that NAM signaling in the thalamus and prefrontal cortical regions consistently reduced mechanical hypersensitivity. However, they observed divergent effects on these measures in the basolateral amygdala. The authors attempted to solve the discrepancy in behavioral measurements between mGlu5 signaling in the basolateral amygdala by determining how NAMs modulate synaptic transmission or in vivo firing and found that these effects were projection-dependent. Overall this paper presents some interesting observations, but the lack of suitable control groups, limited behavior tests, and the broad profiling approach lacks some cohesion and makes it challenging to determine where and how mGlu5 receptors are exerting these effects.The behavior and slice physiology experiments are both lacking control groups. Given the differences in basal and mGlu5 PAM stimulated InsP levels in figure 1, including sham groups for the behavioral testing could provide further insight into how mGlu5 NAMs influence nociception. At a minimum, at least one vehicle-treated and photostimulated group should be included to rule out potential off-target effects on the contralateral brain region to the nerve injury. This could also help address whether mGlu5 NAMs are directly analgesic or anti-nociceptive.

Thank you for this observation. We added all control data with inactive and light‐activated JF‐NP26 in sham‐operated mice in the new supplementary Figure S3. No experiments with alloswitch‐1 were performed to minimize the number of animals used in this study.

Does photostimulation or systemic NAM injection impact other behaviors outside of mechanical withdrawal thresholds? This is important for the interpretation of these effects as pro/anti-nociceptive and not due to impacting vigilance, fear, anxiety, etc. which can affect these sensory measurements.

Thank you for this question. Please, see response to reviewer #1. Additional assays for depression‐ and anxiety‐like behaviors are now included.

I appreciate the effort to link the differences in behavioral responses with NAM activation in the amygdala using slice physiology. In the proposed models, mGlu5 is proposed to exist on some upstream projections. However, no direct effects were observed from these proposed inhibitory (Figure 3a-h) or excitatory (Figure i-p) populations, arguing against this. In the mPFC model in Figure 3, the authors are measuring a monosynaptic EPSC (presumably) from the BLA. If this model is correct, wouldn't this excitatory input also be enhanced onto the proposed mGlu5+ inhibitory neurons leading to an increase in polysynaptic IPSC amplitudes? Similarly, if the NAM is reducing mGlu5 tone on the excitatory BLA projections to feed-forward inhibitory circuits in the CeA, wouldn't a reduction in monosynaptic EPSC amplitudes also be observed?

We greatly appreciate the reviewer’s comment and questions about the proposed circuitry model. We agree that our previous diagram needed clarification and have updated Figure 3a (and summary Figure 4) accordingly. In the mPFC, mGlu5 would act on interneurons that modulate excitatory input (presumably from BLA) presynaptically for decreased excitatory drive of pyramidal cells. Thus, blockade of mGlu5 with alloswitch would remove the inhibitory influence and shift towards excitatory drive of pyramidal cells, which is what the data show (increased EPSCs). Regarding the BLA‐CeA synapse, our data suggest that mGlu5 preferentially acts through the feedforward inhibition pathway rather than on the monosynaptic excitatory drive of CeA output neurons. It is possible that different synaptic signaling mechanisms are involved at interneuron versus output neuron synapses.

Related to the point above, does Alloswitch-1 modulate membrane properties of the recorded output neurons and might there also be a direct effect on these microcircuits?

Thank you for raising this interesting point. The focus of the brain slice physiology was on the synaptic circuitry and feedforward inhibition in particular. Given the time‐sensitivity protocol, which involved four different manipulations (predrug, drug, and two different light administrations), we did not include a thorough analysis of membrane properties and excitability. However, we did not overserve any changes in holding current or conductance in the voltage‐clamp experiments.

What light intensities were used, and does violet illumination affect cellular health?

Please see our response to reviewer 2. The light intensities used in our experiments were carefully chosen based on previous studies to ensure effective isomerization of the photoswitchable ligands while minimizing potential phototoxicity or cellular damage (Pittolo et al.,Nat. Chem. Biol., 2014; Gómez‐Santacana et al. ACS Cent. Sci., Ricart‐Ortega et al., ACS Pharmacol. Transl. Sci. 2020). Regarding the potential effects of illumination on cellular health, we took this concern into consideration for our experimental design. We selected 405 nm, which is fully located in the visible spectrum (violet range) and is less energetic and less toxic than UV‐light. The extent of alloswitch‐1 photoisomerization from trans‐to‐cis configuration is comparable for illumination with 380 or 405 nm. Still, violet light, particularly at high intensities and prolonged exposure durations, may induce phototoxicity and affect cellular viability. To mitigate this risk and minimize any adverse effects on cellular health, we employed pulsed light delivery protocols, as pulsed light illumination allows for precise control over the duration and intensity of light exposure, reducing the risk of phototoxicity while ensuring effective isomerization of the photoswitchable ligands. By using pulsed light, we aimed to strike a balance between achieving optimal experimental outcomes and safeguarding cellular viability.